# Indigenous Peoples, Exclusion and Precarious Work: Design of Strategies to Address Poverty in Indigenous and Peasant Populations in Ecuador through the SWOT-AHP Methodology

**DOI:** 10.3390/ijerph18020570

**Published:** 2021-01-12

**Authors:** Jorge E. García Guerrero, Ramón Rueda López, Arturo Luque González, Nuria Ceular-Villamandos

**Affiliations:** 1Department of Statistic, Econometrics, Operational Research, Business Organisation and Applied Economics, University of Cordoba, 14071 Córdoba, Spain; z82gaguj@uco.es (J.E.G.G.); nuria.ceular@uco.es (N.C.-V.); 2Faculty of Social Sciences, Technical University of Manabi, 130105 Portoviejo, Ecuador; arturo.luque@utm.edu.ec; 3School of Administration-Research Group in Direction and Management, University of Rosario, 111711 Bogotá, Colombia

**Keywords:** public strategies, poverty, indigenous peoples, indigenous women, ethnic minorities, decent work, social and solidarity economy

## Abstract

This research analysed the options that, following decent employment and the social economy, can allow the human development of poor, excluded and vulnerable indigenous populations in Ecuador. A set of strategies were developed which can be implemented by public authorities and by community organisations. They were designed from two types of expert consultations: the Delphi method and the analysis of Strengths, Weaknesses, Opportunities and Threats (SWOT) combined with Analytic Hierarchy Process method (AHP) for hierarchizing the criteria collected and obtaining strategies. The proposed strategies are as follows: adopting appropriate legal frameworks, respecting peoples’ rights, better distribution of public resources, implementing monitoring systems, developing solidarity markets and recognizing the participation of the poor as a subject of rights. This investigation revealed differences between the state, which identifies the poor with monetary indicators, and the indigenous peoples, who see it as the lack of community links, by conceiving the poor as a beneficiary of official assistance, despite the fact that a strong community and peasant organisation could be used. The value of an economy based on reciprocity and confidence was also recognized, identifying niches of production and consumption to create partnerships and ensure the participation of indigenous peoples in decision-making areas.

## 1. Introduction

The livelihoods and occupations of indigenous peoples are deeply related to land and natural resources in a rural and community economy [1] where they are involved in traditional occupations and activities [2]. The formulation of coherent and responsive public policies does not include them within their participation and consultation processes by hiding them in official data and studies, even though people could provide solutions to achieve the 17 Sustainable Development Goals (SDGs) of the 2030 Agenda or in the construction of a future of work, sustainability and inclusiveness [2].

The human capital of indigenous peoples is forgotten at national levels [3], and educational achievement, in particular higher education, remains pending and traditional knowledge systems are not used [4]; thus, formal education or employment systems do not lead to social mobility or socio-economic empowerment [5]. The reality of indigenous women, who claim equal opportunities [6], or hard criticize the hegemony of gender inequity in small communities [7], has not been adequately addressed by scientific literature, and they also continue to be invisible in official data; their aspirations are rarely part of the discussion on policies affecting them [8].

Indigenous experiences are intertwined by social and material exclusion, breaking family and communal ties, loss of cultural identity, intergenerational trauma, in urban areas are precarious and exclusive, with lack of access to health, housing (including social housing), well-being, use of public space and political representation [9].

In a report issued by the International Labour Organization (ILO), climate change, land degradation and natural resource exploitation are putting pressure on the livelihoods and traditional occupations of indigenous peoples [10]. Rural residents are ignored by the allocation of government resources, for which there is no adequate accountability, while decentralization and urbanization, including collective territories, are part of State control over rural resources and land [11]. The gaps in indigenous and non-indigenous life expectancies are associated with the conditions of poverty that many are suffering [12].

The continuing search for economic opportunities causes the migration of indigenous women and men outside their traditional territories, in some cases, to work in the formal economy and to create enterprises, including cooperatives [13], and in others, to face greater dependence on occasional work in agriculture, construction, domestic work or informal trade [1] where they attain their livelihoods and as a result of the lack of opportunities in the formal economy [14].

Inequalities in the world of work prevent the end of poverty of indigenous peoples. In Ecuador, the status of indigenous people plus that of inhabiting a rural area means multiplying by three the percentage of women who are poor [15]. This is due to a higher rate of informality among indigenous people with work, overrepresentation in the rural population, lack of land ownership, lack of access to natural and productive resources and poor access to education, care and infrastructure services [16]. Breaking the vicious circle of poverty and poor health is done partly by associating health sector activities with other sectors and with the use of human rights standards and guidelines [17].

Studies, such as Vazquez-Maguirre and Portales [18], conclude that indigenous social enterprises are capable of generating well-being in communities where they operate and reducing exclusion patterns; Jaramillo and Jácome [19] express that the transition from popular to popular economies explains how an urban indigenous population, from associative forms, manage to satisfy their need for legitimate goods and services in decent terms; in addition, for Hussain and Endut [20], the social dialogue, stability and security of an enterprise have an impact on reducing the conditions of employment and on the balance between the personal and working life of small businessmen and self-employed entrepreneurs.

In view of the above, the theoretical objective which this research aims to achieve is to identify, from a decent employment perspective, and in particular from an economy that recognizes justice as a value in exchanges, alternatives to the development of populations affected by poverty and exclusion, especially indigenous and rural.

On the other hand, this theoretical objective leads to the practical objective of developing strategies that, based on the social economy and solidarity, promote decent employment and market access as useful mechanisms for overcoming poverty and inequality, especially for indigenous women’s populations in rural areas of Ecuador, making it possible to improve their well-being and living conditions.

The analysis of information collected, through consultations with experts (Delphi and SWOT-AHP), followed the five-step scheme shown in Figure 1, which part of the identification of analysis dimensions for social and solidarity economy (SSE) as a fostering of well-being in the study populations, which is an objective in this research.

On the basis of the analysis dimensions, open questions were designed within the Delphi consultation, to establish the lines of the experts linking the SSE and poverty alleviation of indigenous peoples, as well as to know what initiatives to consider within the actions that are, in a later manner, with these criteria building the SWOT matrix, which, using the AHP method, will make it possible to prioritize criteria and establish strategies that, based on SSE, promote decent employment and market access as useful mechanisms for overcoming poverty and inequality.

This research required the selection of a suitable tool for multi-criterion decision-making, which is why the AHP method developed by Saaty was chosen [21]. This method, which is used to facilitate the decision on multiple criteria, allows for the order of the problem or objective on which to decide, in a formal hierarchical structure, with criteria, sub-criteria and alternatives for decision which, as a whole, constitute decision factors [22] which in this particular case mean options such as: ensuring individual and collective rights of peoples, develop management capacities in their economic organisations, establish appropriate policies to strengthen SSE and strengthen community organisations. With these options, the study pays the debate on the effectiveness of SSE to generate well-being. Taking that discussion, in subsequent studies, towards the reality of rural women of indigenous peoples settled in Tungurahua province in Ecuador.

This research deals with the analysis of SSE, decent work and its relationship to the reality of indigenous peoples, especially women, by putting the dimensions of analysing consultations with experts at the centre of the research methodology. Following the analysis of the data, practical results of the investigation are achieved, while the final conclusions address the theoretical objective.

## 2. Literature Review

Theoretical references of the research required revision of the scientific literature on social and solidarity economy, indigenous peoples, poverty and inequality. This review was carried out using two of the main scientific bases of reference: Web of Science (WoS) and Scopus. This search was carried out using key criteria for research such as: “social economy”, “poverty”, “indigenous”, “indigenous women” and their relationship to other terms such as “inequality”, “exclusion”, “reciprocity” and “community”; secondarily, Google Scholar was used.

The time framework was over the last five years, both in English and Spanish, however, with the intention to incorporate those sources considered as reference for research, in some cases the time framework was extended, especially in the case of the argument of the Delphi, SWOT and AHP methodologies used, a time framework was not established on those issues.

### 2.1. Social and Solidarity Economy

There is a wide range of concepts around the notion of social economy, including labour economy, cooperative economy, associative enterprises, and family and non-family micro-enterprises, which share the basis for work rather than capital as a factor that organises and runs the organisation [23]. The solidarity economy, which is born from the common log of the social economy, its principles include promoting both production, distribution, consumption and financing relations based on values such as social justice, and cooperation on reciprocity and mutual aid [24].

For Coraggio [25], the popular economy, or the social economy, is the aggregate sector of the household units of workers and their ad hoc organisations, which consists of: (1) the set of subjective and material resources, private and public, which command group or unit domestics (single or collective, family or community) depending on their continuing performance on their working fund; (2) activities undertaken to meet their needs immediately or not through self-dependent, commercial or non-commercial, competitive or cooperative activities; (3) the habits, rules, values and knowledge that guide such activities; (4) the corresponding groupings, networks and relationships—of concurrence, regulation or cooperation, internal or external—that they institute through the formal organisation or the repetition of these activities.

From this definition, it is possible to draw the four dimensions of analysis contained in Table 1 and to link them to open questions included in the first consultation of experts (i.e., the Delphi method).

In the face of capital and its accumulation, the solidarity economy puts people and their work at the centre of the economic system. Access to products from work is not only possible through monetary income, it is possible through trade based on different types of reciprocity or through trade in products and services [23], leaving an instrumental role for the market in the welfare of people and the reproduction of life on the planet [24].

Social economy and solidarity are able to distribute more fairly by considering the pre-eminence of work on people’s financial capacity, seeking the provision of goods and services necessary for society based on self-organized work and environmental sustainability, acting at local and comparatively small level in proximity to economic circuits [34]. It seeks to produce and distribute sufficient material wealth so that it can generate sustainable conditions of self-managed development for each individual, societies and the planet itself [35]. In analysing solidarity entrepreneurship, Arboleda and Zabala [30] concluded that success cases are related to the proximity of their social object, their ability to take root in a particular territory, operating as a process of chains of companies of the same type and to encourage self-management and participatory processes. Cooperativism is not an ideology but a solidarity system with the capacity to strengthen sustainable development processes [36].

### 2.2. Decent Work

Decent work is the justly paid productive occupation that is exercised in conditions of freedom, equity, security and respect for human dignity [37]. The decent work programme contains four strategic objectives of equal importance: promotion of employment, social protection, social dialogue and tripartism, and fundamental principles and rights at work, and it incorporates gender equality and non-discrimination in a cross-cutting manner [38]. Decent work is a social innovation established by the ILO to address the impact of globalization on employment, labour rights and conditions, occupational safety and health, social protection and social dialogue [39].

The labour market situation in Ecuador, with general levels of employment in the informal sector, underemployment and unemployment as at December 2019, 46.7%, 17.8% and 3.8% respectively [40], are aggravated for indigenous people without formal figures, with multiple causes, some shared among indigenous and non-indigenous peoples, such as the lagging in the agricultural sector, mini-founding agriculture, rural migration to the city, low income or ethnic marginalization, lagging behind a dominant colonialism culture, among others [41].

The Constitution of the Republic of Ecuador promulgated in 2008 defines work as a right and social duty, poses a decent and stable economic system, in addition, for the fulfilment of the right to work of communities, peoples and nationalities, proposes specific measures to eliminate discrimination, recognize the forms of organization of the work of peoples, and ensure equal access to employment [32].

The 2030 Agenda for Sustainable Development covers economic, social and environmental sustainability and includes, together with decent work and economic growth (SDG 8), the end of poverty (SDG 1), gender equality (SDG 5) and reducing inequalities (SDG 10), among others [42]. In turn, with SDG 5 and SDG 8, it shows that female employees do not double the addition of domestic and care tasks without considering support for the State’s social infrastructure, redistribution of gender roles and recognition of domestic work [43].

Despite employment, for almost one-third of people in extreme and moderate poverty, this is inadequate or unpaid employment, low qualification and without social protection [44].

Precarious employment arises with unemployment, particularly young people; with informal employment without labour rights and underemployment caused by temporary contracts, low-wages or low-wage hours [45].

For Miguélez [46], the indefinite search for competitiveness leads to the flexibility of staff, with unsafe workers in their employment and unable to raise demands, also weakened as a collective subject, with the argument that these flexibilities meet market needs or alleged advantages for companies but end up affecting household income, which is the result of what Falguera [47] called flexi-security. By withdrawing the person from the right to decent work, the socializing factor is removed from him and, in the face of this lack, there is little left: disease, violence and suffering [48].

### 2.3. Social Economy and Decent Work

Alternatives to development, such as good living, are born from peasant and indigenous approaches inserted into local forms of production [49] and the idea of a sacred nature and principles, values and communal coexistence practices [50]. Collective effort, solidarity, reciprocity, strong family ties, intense work, harmony and respect for nature are traditional cultural values of indigenous peoples [51]. For Padilla-Meléndez [27], the associative values of indigenous cultures promote business, although most of them in need, of indigenous entrepreneurs who then base their activities on individualism, assigning the family the role of recovering community values.

Many demands are being met by the social economy and, thus leading to the creation of new jobs and with them greater well-being, even though those generated, according to Miguélez [52], demand cultural and institutional changes and the incorporation of better technology and greater complexity into their processes.

Solidarity organisations promote the construction of a fairer society and better shape of democracy, politics and economy [53] and contribute significantly to achieving SDG 2, SDG 5, SDG 8, SDG 12 [30] as well as SDG 10 [54].

In social economy organisations, according to Castro Nunez et al. [54], there is greater participation of women, more stable and resilient employment, better access for disadvantaged groups who are socially excluded from good-quality jobs. In the urban economy, women, including indigenous women, represent greater autonomy and, in some cases, economic independence [2].

## 3. Materials and Methods

### 3.1. The Delphi Method

The Delphi technique is a method of structuring a group of individuals, such as a whole, to deal with a complex problem [55], and which contains a credible group of experts [56], and through the consensus of its responses [57], inferences on newly subjective data can be made.

For Landeta et al. [56], the Delphi method is capable of proposing solutions to complex problems, while Del Rincón et al. [58] define it as a specific type of group interview and Uhl [59] as a process of communication between several subjects, through successive questionnaires to reach consensus without personal interaction. With the Delphi method, according to Dalkey [60], anonymity, controlled feedback and the group’s statistical consensus remains.

The Delphi technique begins with the delimitation of the problem, the construction of a first questionnaire and the selection of subjects [57]. The expert panel will be provided with a first consultation with open questions for, using their responses, to develop a second questionnaire with closed options, which will allow statistical treatment in consensus building. After this, further consultations are referred to each participant, including the average score allocated by the group to each questioning together with the expert’s individual qualification, so that they may maintain or modify their response and include the grounds for their decision.

The size of the panel, which depends on the objectives and conditions of each study, should allow a diversity of views [61] without affecting the viability of the process itself. Olabuenaga and Ispizua [62] indicate that the panel should comprise 10 to 30 experts representing different positions on the problem [57] without being a statistical sample of a population.

The sending of new consultations is an iterative process that ends when a high consensus is observed among participants or when it is considered that such consensus will not be increased by subsequent consultation rounds [61]. Authors, such as Olabuenaga and Ispizua [62] and Fernández Ballesteros [61], point out that the first questionnaire will be made up of open questions, while other authors, such as Pérez-Campanero [63], consider it possible to delete the first questionnaire and to initiate proceedings directly with closed questions when the problem has been defined earlier and precisely.

In order to consider the existence of consensus, in the case of climate questions, such as this study, it was feasible to use the percentage reached in one or some of the categories of response [57] for that purpose; the criteria for the statistician that will be considered acceptable should be anticipated, excluding from the following consultations any questions that meet that criterion.

### 3.2. SWOT-AHP Methodology

The Strengths, Weaknesses, Opportunities and Threats analysis (SWOT) has been very acceptable since Learned et al. [64] defined this methodology. This methodology explores the impacts that different forces may have on a particular system. The purpose of the methodology is to identify, synthesize and assess external factors measured in threat and opportunities, and those factors within the meaning of weaknesses and strengths, which have a positive or negative impact on a given in the system under review [65].

However, SWOT analysis is a qualitative method that does not allow direct prioritisation of strategies and should be combined with other methods that incorporate alternatives to make decisions based on multiple criteria [66], which consider both qualitative and quantitative factors and leads to an optimal solution [67]. The multi-criterion analysis methods include approaches which, according to Tscheikner-Gratl et al. [68], can be grouped into three general categories:(1)By approach, which establishes a preferential relationship among alternatives to determine the most dominant ones.(2)By the target approach, aspiration and reference level, where the distance between alternatives and specific solutions is measured and alternatives are recognised close to the ideal solution.(3)By the value measurement approach, where the weight of each factor is determined when comparing pairs of criteria, which are then given a score for its priority; this approach belongs the method of AHP (analytical hierarchy process).

As a general rule, the multi-criterion method should be selected according to the objectives of the decision-making problem. In this particular case, it was considered: variation in the degrees of criteria; independence of criteria; subjective preferences of decision-makers; in addition to the number of criteria and sub-criteria incorporated [69], conditions under which the AHP method, which is one of the methods widely used in the Multiple Criteria Analysis [70,71]

The AHP method proposes a way to order analytical thinking according to three principles: building hierarchy, prioritisation and logical coherence [72]. The AHP methodology requires a hierarchically structured problem for which a comparison of factors can be defined [73]. The difference between HPA and other decision-making methods is that during the process it provides a comprehensive structure to combine intuitive and rational value; and, as well as verifies consistency in decision-making [74].

The technique, developed by Saaty, allows the reality perceived by the individual to be moved to a scale of reason for determining, through a mathematical peer comparison architecture, which, among several, is the best possible alternative or decision, facilitating strategic planning and decision making under multiple criteria [21].

The relevance of each factor is determined by comparing elements two to two in consultation with experts to select the most relevant (A or B) and then assess them by a significant scale (between 1 and 9) (see Table 2), in the end, the consistency of the scores allocated with the consistency index or ratio [75].

The model formulated is assessed by the confidence that it can achieve, this confidence is measured through the degree of consistency of the model. This consists of the possibility of measuring, to what extent, comparisons of factors made may be considered stable. The consistency of the model is measured through the consistency index or ratio (Cr). The Cr is calculated for each matrix of comparations. Its value should be below 0.1, which reflects a good adjustment or good stability of the model. However, the value of the consistency index does not need to reach value 0. In this regard, a value of the consistency index between 0 and 0.1 represents how far a better understanding of the problem can be achieved [76,77].

In conjunction with the SWOT matrix together with the AHP method, it provides a qualitative reference framework for the analysis of external and internal factors affecting a decision, from which a quantitative assessment is incorporated in order to assess and weigh in a systematic and understandable manner each of these factors and compare its hierarchical position and intensity [79]. With the implementation of the AHP model, it is possible to determine, by means of a mathematical architecture, the overall level, the best alternative or possible decision-making, by facilitating decision making under multiple criteria and strategic planning [21].

The SWOT-AHP methodology has been implemented in several sectors [80], for example, industry [81], energy [82], agriculture [83], tourism [84], engineering [85] or territorial development [86].

All experts who participated both in the Delphi stage and the SWOT-AHP stage were their role, objectives and scope of the investigation, while requesting their consent before the consultation rounds began.

### 3.3. Definition of Strategies

The strategic matrix proposed by Weihrich [87] will be used for the definition of strategies. This matrix, presented in Table 3, contains four groups of strategies that emerge from crossing external and internal factors results in the SWOT analysis: weaknesses–threats (WT), weaknesses–opportunities (WO), strengths–threats (ST) and, finally, strengths–opportunity (SO). Prioritising those subfactors that have become more relevant in the previous phase, it is generally at this last phase of maximising strengths and opportunities and minimising weaknesses and threats.

Thus, SO strategies aim to maximise both strengths and opportunities, while ST strategies are based on the strengths that can face threats in the environment. The WT strategies are created by minimising both weaknesses and threats, while the WO strategies seek to minimise weaknesses and maximise opportunities [87].

## 4. Analysis and Discussion

### 4.1. Application of the Delphi Method

#### 4.1.1. Construction of Questionnaires

The first questionnaire consisted of two blocks of questions. The first was composed of seven open questions: (1) on official speech and its impact on the real decline in poverty; (2) on the ability of organisations and states to respond to new forms of work; (3) on the space of SSE and community economies in local and international trade; (4) on state actions, enterprises and households to change their exclusion conditions; (5) on relations between production and consumption with conditions of equity within countries; (6) on the impact of social demands on Ecuador and Latin America; (7) on the impact of 2030 Agenda for Sustainable Development. The questions were built in relation to the dimensions of analysis as shown in Table 1.

The second block contains forty closed questions and includes several answers to the objectives of this investigation, and which were defined in advance to avoid a first consultation of questions opened exclusively. The analysis took the answers to the seven open questions and nine closed questions relating to the social economy and solidarity. Each of these questions was answered by a Likert-type measurement scale, common use in Delphi [88], to which scores between 1 (total disagreement) and 7 (total agreement).

#### 4.1.2. Expert Panel Delphi

For the selection of the panel of experts, it was considered that it (1) include representatives of indigenous peoples of Ecuador and Latin America, who occupy or have occupied leadership areas within their organisations or communities; (2) experts on poverty, inequality, gender equity, community economies, social economies and solidarity or human rights; (3) Ecuadorians with experience in public social policies, including those related to SSE and poverty eradication; (4) that the panel reflects the equity of women and men in its composition. Thus, invitations were convened and sent to 40 people who met the defined criteria, 16 of whom consented in the first round of consultations, and the panel was formed as shown in Table 4.

#### 4.1.3. Consultation Rounds

Two rounds were considered to avoid termination of the expert panel and because the first round obtained 52.5% of items that met the consensus criteria, i.e., responses grouped in three continuous values reaching 75% of the total panel responses.

For closed-type questions, the evaluation used central trend and dispersion measures [89], in particular, median and interquartile distance, data that together indicate the position and dispersion of the set of answers for each item, taking into account their distribution [90].

The second round of consultations considered consensus and the stability of responses, analysing, in addition to the concentration of response values, the interquartile distance and the variation of the replies between rounds. The consensus was thus obtained in 47.4% of the remaining items, in total, 75% of those initially proposed, see Table 5.

In order to measure the group stability of successive rounds, the variation rate of relative interquartile routes [91], calculated as the difference between upper (Q3) and lower (Q1) quartiles divided by the median, was chosen. The iterative process seeks values between −0.25 and 0.25 in the variation of the relative interquartile route between the first and second round. The final result was stability in 85% of the total items consulted, together with 68.8% participation of the panel of experts in the second round, where it was decided that a new round was neither necessary nor feasible due to the danger of further reduction in the sample size.

For the analysis of Delphi results, the responses of each expert [92] were considered equivalent, without differentiating or weighting them under any criterion, since studies such as Sackman [93] reported absence of a correlation between expert intelligence and the accuracy of their estimates.

Agreements obtained in the consultation, Table 6, relating to the generation of employment in SSE for indigenous populations, include the validity of SSE for improving the lives of indigenous women, the relevance of relationships of confidence and reciprocity, the importance of work with adequate and dignified characteristics and the importance of training in adolescent populations and children.

### 4.2. Application of the SWOT-AHP Methodology

Analysis of strengths, weaknesses, opportunities and threats: development of the SWOT matrix.

For the construction of the SWOT matrix in Table 7, the responses collected in Delphi were based on the responses collected as strengths (S), weaknesses (W), opportunities (O) and threats (T), and based on the four dimensions of analysis in Table 1: individual or community relations; management capacity of organisations; integration of the popular system and solidarity; and, at the central and territorial levels, rules, institutions and action.

#### 4.2.1. Development of the Hierarchy of Decisions

The aim of presenting the hierarchical dependency relationships between criteria and decision sub-criteria with strategic decisions is presented in Figure 2, where four levels were found.

The first was on which strategies for strengthening SSE in Ecuador should be established; the second level was the decision criteria, in this case strengths, weaknesses, opportunities and threats; the third level consisted of the decision sub-criteria that were consistent with each and every element that consisted of the outcome of the SWOT analysis; leaving for the fourth level, strategic decisions resulting from the analysis of the SWOT matrix.

#### 4.2.2. Measurement of the Relevance of Criteria and Sub-Criteria through the AHP Methodology

A further consultation of experts was raised for measuring the relevance of the criteria, generating a specific sample to which the AHP methodology was applied. The experts responded to a peer comparison SWOT form and sub-criteria contained in Table 7 to establish those of the greatest relevance in the opinion of the latter. This method does not require a meeting at a given location, does not allow the interaction of experts, thereby bringing great value in avoiding biases. Each expert was asked to provide a direct estimate of his approach and his responses as a whole were mathematically treated. The algorithmic treatment of the values offered by the experts and the calculation of each consistency ratio has been performed through AHP Excel template with multiple inputs [94].

For the consultation of experts, 11 invitations were sent to managers, academics and public officials related to the People’s System and solidarity in Ecuador, with five positive responses and consequently five filled forms, the results of which are shown in Table 8 and Table 9.

According to the expert criteria, strengths (38.1%) and opportunities (29.7%) had the greater weight than weaknesses and threats at the time of analysing the SSE in Ecuador. In the first group, the fortresses, as set out in Table 8, highlights the F4 that proposes a community economy focusing on the needs of the population and sustained on their own resources; whereas, in the second group, the major opportunity was the O5 which proposes to recognise the poor as an actor and political subject in decisions affecting him. This suggests that there are potential strengths and opportunities that are not used or are not put into value by decision makers.

### 4.3. Definition of Strategies

As a result of the SWOT-AHP analysis, four sets of strategies are being raised (Table 10), grouped as weaknesses–threats (WT) or mini–mini strategies, weaknesses–opportunities (WO) or mini–maxi strategies, strengths–threats (ST) or maxi–mini strategies and, finally, strengths–opportunity (SO) or maxi–maxi strategies [78].

The hierarchisation of criteria emerges from the need to establish strategies focused on strengthening strengths (relevance: 38.09%) using opportunities (relevance: 29.71%); in particular, S4 (2nd), S2 (5th), S5 (6th), S3 (7th), S6 (10th), with opportunities O5 (1), O3 (4th), O4 (9th); as well as those of turning W6 (3rd) and W5 (8th), into opportunities.

As shown in Table 10, to meet the first outcome of hierarchisation, the strategy group SO was defined: strengthening organisations, subjects and actions that have the capacity to build a self-centred community economy with a focus on the needs of the population (SO1); the aim of the new economy models recognising the community’s organisation and economy (SO2); identifying niches of production and consumption (SO3); guiding new forms of work, respect for indigenous peoples’ cosmovision and respect for human rights (SO4); and designing national public policies and establishing international treaties to support the social economy and the community economy (SO5).

For the second, the development of weaknesses into opportunities was a three strategies WO: including the poor, as a political subject and subject of rights in policy-making processes (WO1); establishing social programmes and maintaining them to adequate levels of education, health and nutrition (WO2); and incorporating into agricultural enterprises, technically and ecologically produced means (WO3). In addition, ST and WT strategies are being developed as shown in Table 10.

## 5. Conclusions

The first theoretical conclusion of research was born from the difference with which the state, which uses monetary indicators, defines poverty and how indigenous peoples recognise it, for those who lack their relationship with the community. The public authorities, by distinguishing the poor from purchasing capacity or consumption, promote assistant programmes in which the populations concerned do not participate. These communities end up invisible within official figures, reduced to beneficiaries of state assistance and disqualified as political subjects or without rights, when, in conceiving poverty from multiple dimensions, they should recognise the need to implement policies with a rights approach and dimensions such as economic standards, living conditions, education, training and health care [95].

A second theoretical conclusion recognises that new forms of work, with the highest technology, relocated and automated, will lead to poverty eradication as soon as they respect the cosmovision of indigenous peoples and their rights, allow them to equally participate in the construction of the information society from their general conceptions to their implementation [26], by forming a knowledge system in which people participate in decision-making processes and, in exchange, practices both products and knowledge [4].

This investigation further concludes that, as a source of opportunities, the social economy is provided as an alternative to achieving the well-being of vulnerable or excluded groups and the general population. Women working in it enjoy a more stable working life, have resilient jobs and lower gender gaps [48]. The social economy, which in Ecuador could use strong community organisation, requires better distribution of public resources, market development and opportunities to expand the number of buyers and bring farmers closer to consumers in the field and the city. Indigenous peoples place their social structures in confidence, reciprocity and cooperation, where harmonious coexistence is promoted between peoples and nature, recognizing ancestral principles, such as the Sumak Kawsay, which are based on community practices that question economic accumulation, such as “minga”, “randi randi”, change hands or “maki mañachi”.

The practical objective conclusions are summarized in five SO strategies to: strengthen organisations with the capacity to build settled community economies on endogenous factors and resources (SO1), articulate models recognised by those organizations and their economies (SO2), identify niches of production and local consumption that pretend social practices (SO3), guide new forms of work towards peoples’ cosmovision (SO4) and design policies or treaties to support popular economies (SO5). Four ST strategies seeking citizens’ participation in matters affecting them (ST1), develop programmes to promote and strengthen rural and community economies (ST2), consolidate networks that enable SSE organisations to compete with more political and economic power (ST3) and recover solidarity and reciprocity practices of peoples and their economic institutions (ST4); three strategies WO incorporating the poor as a political subject (WO1), demand the continued social programmes to reach sufficient levels of education, health and nutrition (WO2), and provide added value for SSE products (WO3); plus four WT strategies aimed at generating adequate employment to avoid poverty and rural migration (WT1), consolidate markets, in short chains not dependent on large trade (WT2), facilitate access to seed, technical assistance, finance and insurance for their production (WT3), as well as ensuring training for employment and access to education, including higher levels in study populations (WT4).

As regards the limitations of this investigation, the methodologies used should be identified. In this regard, both Delphi and SWOT-AHP depend on the contribution of experts consulted and are exposed to delays or the abandonment of participants during iterations. It is important to ensure the selection of experts to ensure the representative of the sample and to avoid its mortality.

The selection of experts includes persons not in Ecuador, and therefore selected Delphi and SWOT-AHP methods, supported by electronic forms and e-mail communication, facilitated the process of information updating, which mostly coincided with the freezing periods set out as measures to address the pandemic by COVID-19.

This research took on the expert criterion, but it omitted, for reasons of design and as part of future research, the vision of the study population, even though recognising the risk of the expert criterion that might be affected by some particular interest or experience.

Future lines of research should include experiences drawn directly from excluded individuals, populations and communities, as well as experiences of adequate and inadequate employment conditions, especially within the informal economy, and those that provide knowledge about the impact of the Community and family economy to maintain or overcome poverty and inequality. The present research sets out the framework for one next on ancestral practices, indigenous uprising, social capital and community organization as useful mechanisms for overcoming poverty and exclusion, seeking the voices of rural women from the four indigenous peoples settled in Tungurahua province in Ecuador.

## Figures and Tables

**Figure 1 ijerph-18-00570-f001:**
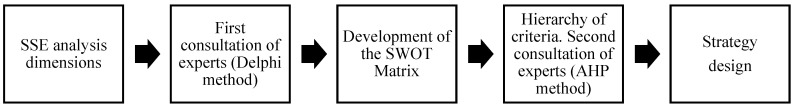
Scheme used for the definition of strategies. Social and solidarity economy.

**Figure 2 ijerph-18-00570-f002:**
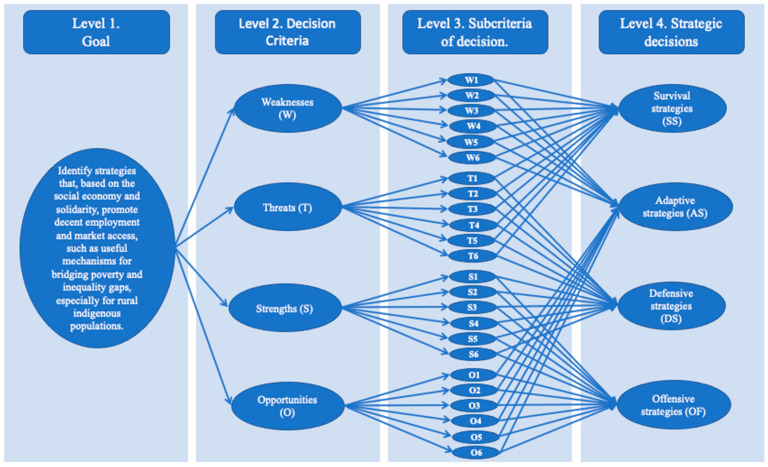
Hierarchical decision model for determining SSE-based strategies for promoting employment and access to product markets.

**Table 1 ijerph-18-00570-t001:** Analytical dimensions.

Variable	Description	Source	Delphi Questions
Individual or Community relations	Based on values, such as solidarity and reciprocity, community organisational and management capacity, networks and relationships based on domestic groups related to moral behavioural patterns, expressing the free will of a common work that generates links of friendship, solidarity and community.	[19,25,26,27,28]	In Latin America (Ecuador, Chile, Colombia, Bolivia, etc.), there have been significant social mobilizations demanding substantial changes in policies implemented by their governments during 2019. Are the uprisings (indigenous), social protests, civil disobedience, passive resistance, etc., valid tools for moving towards greater equity and development? What are the weaknesses of the 2030 Agenda for Sustainable Development?
Management of the organisation	Set of resources, capacities and activities which, organised through community forms or self-managed associations, according to principles of solidarity, are useful for the production, distribution, movement, financing and consumption of goods and services. Marketing measures which (seemingly unplanned and informal) are aimed at competition and the market. Participation in decision-making areas.	[4,25,28,29,30,31]	3. How will new working conditions based on automation, relocation, disruption and consumption processes for agriculture populations affect many of them? Are increases in migration or poverty evident? 4. The current model of international trade in production and consumption (globalisation, transnational corporations, free trade treaties).
Integration of the People’s System and Solidarity	Union of efforts between organisations and other business forms of a similar nature or by territorial or productive relationship, to form networks that allow collective maximisation of factors and their ability to succeed, above the experience of cyclical crisis, which is the basis of capital and management factors.	[25,30,32,33]	5. Ten of the large multinational societies control 85% of world food trade. In this context, does the social economy and solidarity processes have a place? Is it possible to articulate a community economy?
State rules, institutions and action at the central and territorial levels	Prevailing legal or moral rules that provide legal and/or correct actions, rights and obligations of citizens and mechanisms for making them effective. Institutions associated with the activities of the social economy.	[25,32]	6. What effective action is there for each State and supranational bodies and which companies will overcome their poverty or exclusion? Do homes have an ability to manoeuvre? 7. Mentioning the poor or poverty within an official speech is usually common for Latin American politicians, an area of real concern for governments to improve the conditions under which the most disadvantaged populations live. Does the poor included in the official (regional and global) narrative translate into real variations in poverty figures? What is its take?

Source: own elaboration.

**Table 2 ijerph-18-00570-t002:** Scale of importance for peer comparison of decision criteria and sub-criteria. Source: Adapted from [78]. Reproduced with permission of Elsevier’s Copyright Clearance Center.

Scale of Importance	Definition	Description
1	Equal importance	Two attributes contribute identically the objective
3	Moderate dominance	Experience or judgment slightly favors one attribute over another
5	Strong dominance	Experience or judgment strongly favors one attribute over another
7	Demonstrated dominance	An attribute’s dominance is demonstrated in practice
9	Extreme dominance	The evidence favoring an attribute over another is affirmed to the highest possible order
2, 4, 6, 8	Intermediate values	Further subdivision or compromise is needed

**Table 3 ijerph-18-00570-t003:** Strategy matrix. Source: Adapted from [87]. Reproduced with permission of EMERALD’s Copyright Clearance Center.

	Threats (T)	Opportunities (O)
**Weaknesses (W)**	“WT-Mini-mini” strategies. These strategies aim to minimize both weaknesses and threats. They are intended to reduce the risk position that may exist.	“WO-Mini-maxi Strategies”. In their case, these strategies aim to minimize weaknesses and maximize opportunities. The aim is to identify the internal weaknesses that put opportunities at risk.
**Strengths (S)**	“ST-maxi-mini” strategies. These strategies aim to maximise those strengths which are to minimise the threats arising from outside.	“FO-Maxi-maxi Strategies”. These strategies aim to maximize those strengths that can make it possible to exploit or develop opportunities for the environment.

**Table 4 ijerph-18-00570-t004:** Conforming the expert panel.

Requirement	Condition	*n*	%
Sex	Men	9	56.3%
	Women	7	43.8%
Origin	Indigenous	5	31.3%
	Non-Indigenous	11	68.8%
Residence	Ecuador	6	37.5%
	Latin America (excluded Ecuador)	6	37.5%
	Europe	4	25.0%

Source: Own elaboration.

**Table 5 ijerph-18-00570-t005:** Results of consultation rounds.

Question	Consensus	Median	Q1	Q3–Q1	Q3
1.1	First round	1.50	1.00	1.00	2.00
1.2	First round	1.00	1.00	0.75	1.75
1.3	First round	7.00	6.00	1.00	7.00
1.4	Round two	5.00	4.25	2.50	6.75
1.5	First round	1.00	1.00	-	1.00
2.1	First round	1.00	1.00	1.00	2.00
2.2	Round two	4.00	1.25	3.75	5.00
2.3	Round two	4.00	4.00	1.75	5.75
2.4	Round two	3.00	2.25	3.50	5.75
2.5	First round	2.00	1.00	2.00	3.00
3.1	Round two	5.00	4.00	2.00	6.00
3.2	First round	5.00	4.00	2.00	6.00
3.3	Round two	4.00	3.00	2.00	5.00
3.4	First round	7.00	7.00	-	7.00
3.5	Round two	4.00	4.00	0.75	4.75
4.1	Round two	5.00	3.00	3.00	6.00
4.2	First round	5.50	4.00	2.00	6.00
4.3	Round two	5.50	5.00	1.00	6.00
4.4	Round two	5.00	4.00	2.75	6.75
4.5	First round	5.00	4.25	2.50	6.75
5.1	First round	6.00	5.00	2.00	7.00
5.2	Round two	3.50	2.25	1.75	4.00
5.3	First round	6.00	6.00	1.00	7.00
5.4	Round two	5.00	4.25	1.75	6.00
5.5	Round two	6.00	4.25	1.75	6.00
6.1	Round two	4.00	2.25	2.75	5.00
6.2	First round	1.00	1.00	1.75	2.75
6.3	First round	5.00	4.00	2.00	6.00
6.4	Round two	5.00	3.25	2.75	6.00
6.5	First round	1.50	1.00	1.75	2.75
7.1	Round two	2.00	2.00	1.50	3.50
7.2	First round	6.00	5.00	2.00	7.00
7.3	First round	1.50	1.00	1.00	2.00
7.4	First round	7.00	7.00	-	7.00
7.5	First round	5.00	4.25	1.50	5.75
8.1	Round two	3.00	2.00	2.00	4.00
8.2	First round	7.00	6.25	0.75	7.00
8.3	Round two	3.00	1.00	5.00	6.00
8.4	First round	5.00	5.00	1.00	6.00
8.5	Round two	5.50	4.00	2.00	6.00

Source: Own elaboration.

**Table 6 ijerph-18-00570-t006:** Agreements reached through the Delphi consultation.

Delphi Question (Closed)	Median	Q1	Q3–Q1	Q3	Interpretation
The social and solidarity economy is a valid option for improving the lives of rural and indigenous women in the Andean countries as well as those of their families.	7.0	7.0	-	7.0	Strong agreement
Political decision makers and public officials usually define public policy without timely or reliable information and without assessing their consequences.	5.5	4.0	2.0	6.0	Strong agreement
In impoverished communities, especially indigenous people, certain social relations and structures, characterized by trust, reciprocity and cooperation among their members, are essential to at least maintaining their living conditions.	7.0	7.0	-	7.0	Strong agreement
Markets are homogenising forces of peoples’ culture.	5.0	4.3	2.5	6.8	Moderate agreement
In certain contexts, informal economy processes comply with labour and ethical requirements, the adjective “informal” is a discriminatory and confiscative construction fuelled by states and administrations.	5.0	4.0	2.0	6.0	Moderate agreement
The services provided by the State (justice, health, education, security) prevail the concentration of services (and enterprises) to the detriment of the rural population.	5.5	5.0	1.0	6.0	Moderate agreement
A young person in a low-income home must work as a support for the family economy, although doing so means dropping out of school.	1.0	1.0	1.0	2.0	Strong disagreement
Low wages, as well as labour flexibility, are needed to improve the competitiveness of and through enterprises to contribute to the development of countries.	2.0	1.0	2.0	3.0	Strong disagreement
Governments and supranational agencies should not impose binding commitments on companies to contribute to the end of poverty and advance human development.	1.5	1.0	1.8	2.8	Strong disagreement

Source: Own elaboration.

**Table 7 ijerph-18-00570-t007:** SWOT Matrix: strengths, weaknesses, opportunities and threats.

Weaknesses (W)	Threats (T)
**W1**. The only beneficiaries are owners of productive export activities, which are usually few and concentrate capital. This creates a particular political economy that reproduces the profitable interests of these groups. **W2**. The State with volatile revenue from income or taxes on such trade fails to address poverty and inequality. **W3**. National production responds to mercantilist schemes of large companies requiring supplies to place on selected captive markets. Production is marked by a lent supply agenda and outside communities. **W4**. Agricultural producers do not own seeds, most of them are patented by industrialised countries. Production costs are expensive, the State does not provide technical assistance, agricultural insurance to cover losses nor soft loans. **W5**. The capacity to place products on the domestic and international market is scarce, production is distributed by a third actor who is the great trader. The poor residents in urban centres are engaged in retail trade in agricultural products from their own rural brothers, in total disadvantage against distributing chains. **W6**. Companies reproduce existing dominance models. As the poor working conditions are deepened, poverty is deepened and migration to urban centres is increasing. In communities there are greater concentrations of older people. Young people go to cities in search of work, and they have convinced them that money is the most important thing. That’s why the Chakras left and then the purchase of collective land by companies.	**T1**. Interests are concentrated on a few families or groups of political and economic power at the national and global levels. Coercive measures are needed to limit the accumulation of wealth. The problem lies in wealth distribution policies. **T2**. Countries make development plans that not all successive governments continue, and there are no long-term public policies. **T3**. The possibilities for access to education, especially those at the higher level, are limited for vulnerable populations. Their absence is transformed into a way to expand poverty belts in cities. Migrant populations are mainly engaged in informal, precarious and intermittent trade activities. **T4**. Country economy dependence on external debt credits and extractive models, mostly over the lives of rural and indigenous populations. **T5**. The growing automation phenomenon is a serious threat to work and, particularly, to people with little human capital, including many of the agricultural workers. **T6**. Governments, executive, legislative and judicial branch have been taken by companies and corruption. They make laws and norms, run the country and are therefore the ones they impose. Citizens cannot make decisions and when they are called for, there are murders of social leaders, there is oppression and submission, and collective property and territories are expropriated. The large majorities do not decide, only in elections where the media, which are of the greatest entrepreneurs, are campaigning for these power groups.
**Strengths (S)**	**Opportunities (O)**
**S1**. Establishing Community economy networks allows alliances that can gradually be extended over some types of monopolies. Long-term challenges which, although far away, are hope for the strong Community organisation and its own regulations that safeguard Community processes, including the Community economy. Social base that has resisted adverse forms of governance. **S2**. The social and solidarity economy aims at local segments that begin to gain value for the current geopolitical and economic uncertainty. The social and solidarity processes are increasing and subsequent to COVID-19, the Community economy will be a basis for recession. **S3**. A community economy that promotes self-focused activity on the needs of the population and sustained on endogenous factors and resources is always open when subjects, organisations and strategies are able to build it. **S4**. Indigenous peoples have forever practised the solidarity economy, which is currently recognised as a popular economy and solidarity. Its economic institutions such as “randi randi” or “minga” are important means for the economic development of peoples. **S5**. The social and solidarity economy, by their very nature, are small in scale and can hardly compete with large international trade. Its role, however, may be relevant in the domestic framework, and in particular in certain vulnerable communities and groups.	**O1**. Consumption is a political act. If consumers have a social conscience, they will tend to buy products with a community economy. To establish the primacy of minimum social rights and ultimately the primacy of the dignity of people. **O2**. If technical and organic production means are incorporated and an aggregated value process of its products is initiated, it will have opportunities to move forward. **O3**. Social programmes established over a regular time to achieve levels of education, health and nutrition are the basis. This will enable us to break the cycle of poverty and access to better development conditions. Supranational agencies can help harmonise regional policies and enterprises from corporate social responsibility, in hand with country development guidelines. **O4**. To articulate new models of economy and thus state. **O5**. A change at how we poverty is needed. To look at it as the result of a process that impoverishes a population will only be reversed when the rules and institutions that perpetuate it are changed. From there, to look at the impoverished not as a recipient of assistant policies, but as a political subject who must participate in redefining those social, economic and political standards and institutions. **O6**. Production and consumption niches are a valid response to the situation of populations in a situation of poor conditions. However, it is difficult to join the global economy.

Source: Own elaboration.

**Table 8 ijerph-18-00570-t008:** Relevance for each decision criterion (SWOT analysis factors) at level 2.

Criteria	Relevance–Importance
Weaknesses (W)	20.09%
Threats (T)	12.11%
Strengths (S)	38.09%
Opportunities (O)	29.71%
Consistency ratio (Cr): 2.62%	

Source: Own elaboration.

**Table 9 ijerph-18-00570-t009:** Relevance of each criterion.

Criteria	Relevance of Criteria at Level 2	Sub-Criteria	Sub-Criteria Assessment at Level 3		Rating Sub-Criteria Relative to the Total	
			Relevance	Ranking	Relevance	Ranking
W	20.09%	W1	5.42%	6	1.09%	21
		W2	6.84%	5	1.37%	20
		W3	11.12%	4	2.23%	17
		W4	15.05%	3	3.02%	15
		W5	24.07%	2	4.84%	8
		W6	37.50%	1	7.53%	3
Consistency ratio (Cr): 6.4%						
T	12.11%	T1	6.33%	5	0.77%	23
		T2	5.42%	6	0.66%	24
		T3	32.16%	2	3.89%	12
		T4	14.99%	3	1.82%	18
		T5	8.79%	4	1.06%	22
		T6	32.31%	1	3.91%	11
Consistency ratio (Cr): 5.5%						
S	38.09%	S1	8.33%	6	3.17%	13
		S2	18.59%	2	7.08%	5
		S3	16.20%	4	6.17%	7
		S4	27.57%	1	10.50%	2
		S5	17.91%	3	6.82%	6
		S6	11.40%	5	4.34%	10
Consistency ratio (Cr): 4.7%						
O	29.71%	O1	9.17%	5	2.72%	16
		O2	5.13%	6	1.52%	19
		O3	24.30%	2	7.22%	4
		O4	15.59%	3	4.63%	9
		O5	35.55%	1	10.56%	1
		O6	10.26%	4	3.05%	14
Consistency ratio (Cr): 9.2%						

Source: Own elaboration.

**Table 10 ijerph-18-00570-t010:** Defined strategies.

Type	No.	Description
Strengths–opportunities (SO) or maxi–maxi strategies	SO1	Strengthen organisations, subjects and actions that have the capacity to build a self-centred community economy focused on the needs of the population and sustained on endogenous factors and resources.
	SO2	Articulate new economic models that recognise the organisation and the community economy.
	SO3	Identify niches of production and consumption; local segments sensitive to human and social rights specific to SSE principles.
	SO4	To guide new forms of work, automation and relocation, respect for indigenous peoples’ cosmovision and respect for human rights.
	SO5	Design national public policies and establish international treaties to support the social economy and the community economy and promote the commitment of enterprises to their workers and their community.
Strengths–threats (ST) or maxi–mini strategies	ST1	Ensure citizen participation in the enactment of laws and norms, avoiding State co-operation by power groups and corruption.
	ST2	To establish programmes to promote and strengthen rural and community economies, especially those of peoples and nationalities.
	ST3	To establish networks and partnerships that by enlargement enable SSE organisations to compete with companies belonging to political and economic power groups.
	ST4	Recover practices of solidarity and reciprocity of nationalities and peoples and their economic institutions, such as the “randi randi” or “minga”, etc.
Weaknesses–opportunities (WO) or mini–maxi strategies	WO1	Including in public policy, the poor, as a political subject, is involved in the definition of social, economic and political standards and institutions affecting him.
	WO2	Establish social programmes and maintain them to reach sufficient levels of education, health and nutrition for the majority of the population.
	WO3	Incorporate agricultural enterprises, technically and organic production methods and initiate a value-added process to their products.
Weaknesses–threats (WT) or mini–mini strategies	WT1	Promote appropriate employment, with appropriate working conditions, avoid poverty or rural migration to urban centres.
	WT2	Consolidating markets, in short chains not dependent on large trade, bringing consumers closer to small traders and producers.
	WT3	Provide programmes to enable agricultural producer access to seeds, technical assistance, finance and insurance for their production.
	WT4	Ensure for vulnerable populations, employment training and access to education, including at the top level.

Source: Own elaboration.

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
