# Peer review of "Indigenous Peoples, Exclusion and Precarious Work: Design of Strategies to Address Poverty in Indigenous and Peasant Populations in Ecuador through the SWOT-AHP Methodology"

_ijerph, 2021, doi:10.3390/ijerph18020570_

Round 1

Reviewer 1 Report

The study is interesting and well-written. 

The abstract of the study is good.

The authors need to elaborate on objective(s) and contribution(s) explicitly in lines 84-108 (page 1-2). 

The theoretical framework of the study is confusing. The author gives the heading Theoretical Framework (Page 3, Line 109) but the literature review is given in the subsections. The authors are supposed to illustrate the theoretical framework of the present study.

The authors have used the SWOT-AHP method for the analysis. A strong logic is required for why the authors used these tools despite the availability of multiple and efficient tools such as fuzzy AHP, TOPSIS, fuzzy TOPSIS, VIKOR, DEMATEL, etc.

However, the results are explained properly and discussed in detail.

Author Response

Dear reviewer,

thank you very much for your comments, no doubt they improve the manuscript, its rigor and depth. Then I will reply one by one to your comments.

  • The authors need to elaborate on objective(s) and contribution(s) explicitly in lines 84-108 (page 1-2). 

In the revised version of the manuscript you will find a new presentation of the objectives and contribution of this research.

  • The theoretical framework of the study is confusing. The author gives the heading Theoretical Framework (Page 3, Line 109) but the literature review is given in the subsections. The authors are supposed to illustrate the theoretical framework of the present study.

We think that renaming this section as Literary Review, it can be clearer

  • The authors have used the SWOT-AHP method for the analysis. A strong logic is required for why the authors used these tools despite the availability of multiple and efficient tools such as fuzzy AHP, TOPSIS, fuzzy TOPSIS, VIKOR, DEMATEL, etc.

You will find in the revised version a more solid argument for why this methodology is used instead of another

Kind Regards and happy new year

Prof. Dr. Ramon Rueda Lopez

Reviewer 2 Report

The statement on page 1 / line 42 "The reality of indigenous women, often discriminated within and outside their communities" is questionable. Unless indigenous women are claiming they are being discriminated against in their communities, it would be biased to make that claim. There are traditional roles according to gender, but who defines something as "discriminatory" is important to note. Quantifying is also questionable; what does "often" consist of and how is it documented? It is also critical that indigenous communities and indigenous peoples are centered in that perspective. It has been extensively documented about indigenous women experiencing discrimination outside their respective communities.

For grammatical clarity regarding attribution, the statement on page 2 / line 46 "According to a report by the International Labour Organisation (ILO)" should read "According to the International Labour Organisation (ILO)" or "In a report published by the International Labour Organisation (ILO)." 

The phrase "indigenous and peasant women" on page 2 / line 59 requires clarification of "peasant." Is "peasant" synonymous with indigenous? Are all indigenous women in Ecuador considered poor, and is "peasant" synonymous with poor, or simply with rural? Please clarify this.

Also, there is an imbalance in this comparison: "In Ecuador, indigenous and peasant women are more likely to be poor and suffer conditions of social exclusion than a non-indigenous man in urban areas." For a balanced comparison, the non-indigenous woman in urban areas should be the counterpart. If exploring sexism / patriarchy in Ecuador is part of the study, it is necessary to make comparisons within a specific group, such as sex / gender. Furthermore, some mention should be made about non-indigenous women in urban and rural areas to provide a clearer context about the politics of prejudice against indigenous women. How different are their realities?

Section 2.2 -- "indecent work" is not defined. Also, there is no mention of illegal work in connection to this population. Are indigenous women likely to be involved in sex work? Are they prohibited from integrating in society? What are the labour laws regarding race/ ethnicity? Are indigenous people considered lower caste? Are their opportunities blocked because of this status? How many "decent" jobs are available to indigenous people, and what types of jobs are they?A

Line 321 -- use "16 of whom" for grammar clarity.

Author Response

Dear reviewer,

thank you very much for your comments, no doubt they improve the manuscript, its rigor and depth. Then I will reply one by one to your comments.

  • The statement on page 1 / line 42 "The reality of indigenous women, often discriminated within and outside their communities" is questionable. Unless indigenous women are claiming they are being discriminated against in their communities, it would be biased to make that claim. There are traditional roles according to gender, but who defines something as "discriminatory" is important to note. Quantifying is also questionable; what does "often" consist of and how is it documented? It is also critical that indigenous communities and indigenous peoples are centered in that perspective. It has been extensively documented about indigenous women experiencing discrimination outside their respective communities.

According to your comment, this expression has been clarified including new bibliographic references

  • For grammatical clarity regarding attribution, the statement on page 2 / line 46 "According to a report by the International Labour Organisation (ILO)" should read "According to the International Labour Organisation (ILO)" or "In a report published by the International Labour Organisation (ILO)." 

has been corrected following your instructions

  • The phrase "indigenous and peasant women" on page 2 / line 59 requires clarification of "peasant." Is "peasant" synonymous with indigenous? Are all indigenous women in Ecuador considered poor, and is "peasant" synonymous with poor, or simply with rural? Please clarify this.

According to your comment, this expression has been clarified

  • Also, there is an imbalance in this comparison: "In Ecuador, indigenous and peasant women are more likely to be poor and suffer conditions of social exclusion than a non-indigenous man in urban areas." For a balanced comparison, the non-indigenous woman in urban areas should be the counterpart. If exploring sexism / patriarchy in Ecuador is part of the study, it is necessary to make comparisons within a specific group, such as sex / gender. Furthermore, some mention should be made about non-indigenous women in urban and rural areas to provide a clearer context about the politics of prejudice against indigenous women. How different are their realities?

According to your comment, this expression has been clarified

  • Section 2.2 -- "indecent work" is not defined. Also, there is no mention of illegal work in connection to this population. Are indigenous women likely to be involved in sex work? Are they prohibited from integrating in society? What are the labour laws regarding race/ ethnicity? Are indigenous people considered lower caste? Are their opportunities blocked because of this status? How many "decent" jobs are available to indigenous people, and what types of jobs are they?

We believe that by eliminating the term "indecent work" from the title of the subsection, the content of the subsection responds more clearly to its purpose.

  • Line 321 -- use "16 of whom" for grammar clarity.

DONE

Kind regards and happy new year

Prof. Dr. Ramón Rueda López 

Reviewer 3 Report

The issue is significant, even more so given the current post-covid 19 pandemic situation in the Americas region and precisely the situation of the indigenous population, where protection strategies or responses have emerged from their communities and based on their cultures. Another aspect to highlights the innovative approach to the structural axes of exclusion of the indigenous population.

However, the methodology defined for this needs a better foundation, that is, why the authors chose this methodological option and why they did not use other qualitative techniques or mixed methods available. The article should be strengthened in the final weaknesses section of the study.

A contribution in the developed framework could be to include in the conclusions section a proposal of the framework that arises from the research presented, presenting it as a theory of change based on the various areas of SO, ST, WO and WT. It would add value to the indicated proposal.

Additional three aspects to clarify:

Specify in the initial description of the bibliographic search for the elaboration of the open questions,  the years and the languages ​​included.

Specify the ethics committee and informed consent of the participants.

Line 337 refers to a possible “bias” in the interaction of the experts. This point is a little confusing in the context of the methodology used. Please, could you explain that in the context of the study objectives?

Author Response

Dear reviewer,

thank you very much for your comments, no doubt they improve the manuscript, its rigor and depth. Then I will reply one by one to your comments.

  • However, the methodology defined for this needs a better foundation, that is, why the authors chose this methodological option and why they did not use other qualitative techniques or mixed methods available. The article should be strengthened in the final weaknesses section of the study.

In the revised version of the manuscript, a more detailed argument for the choice of the proposed methodology has been included,

  • A contribution in the developed framework could be to include in the conclusions section a proposal of the framework that arises from the research presented, presenting it as a theory of change based on the various areas of SO, ST, WO and WT. It would add value to the indicated proposal.

the text included in the conclusions as indicated, tries to respond to your comments, thus achieving what you propose

  • Specify in the initial description of the bibliographic search for the elaboration of the open questions,  the years and the languages ​​included.

These aspects have been included and expanded in the new manuscript

  • Specify the ethics committee and informed consent of the participants.

A paragraph has been included that explains this need in the research.

  • Line 337 refers to a possible “bias” in the interaction of the experts. This point is a little confusing in the context of the methodology used. Please, could you explain that in the context of the study objectives?

This bias has been explained in the revised version in more detail, thus clarifying the effect on research.

Kind regards and happy new year

Prof. Dr. Ramon Rueda Lopez

Reviewer 4 Report

Dear authors,

Thank you for giving me the opportunity to read and review this manuscript.

This is a very interesting article. I have enjoyed reading this paper and I see promise in its ability to provide a relevant study about the topic. However, I think that some genuine amendments must be made before it is ready for publication. I suggest the following changes, in order to be finally accepted:

- The scope of the paper is too wide. It is necessary to focus the topic.

- Introduction: The objective is not clear and it does not refer to Ecuador.

- Line 97: “In this case, in search of better alternatives, the AHP method developed by Thomas Saaty [18].” This sentence seems to be incomplete or their meaning is not really clear.

- It would be also desirable to delve into the theoretical aspects. Section 2 (Theoretical framework) does not address the subject in depth.

- Line 110 to 115 explain the methodology used to study the state of the art. Two comments: a) According to the results, this process was not exhaustive enough, as it is said previously; b) It is not very usual to explain the methodology of consulting the database of bibliographic references used to study the theoretical framework, since it is common knowledge to all scientific work.

- Line 116 to 118: Why it is analyzed only the literature of the last five years? In spite of the fact that it is important to include current references, to encompass the state of the art, time framework should not be delimited. Especially, since it is later said that this time framework is exceeded on some occasions, without the reason being sufficiently justified.   

- Although the paper cites 86 references, wich shows a vast knowledge of the topic, there are too many references to web pages and some references are a little general and not so important to your research. It could also be deepened in other relevant English-speaking references that are missing, according to what it was said before.

Author Response

Dear reviewer,

thank you very much for your comments, no doubt they improve the manuscript, its rigor and depth. Then I will reply one by one to your comments.

  • The scope of the paper is too wide. It is necessary to focus the topic.

In the revised version of the manuscript, aspects related to the geographic frame of reference have been included with which to specify and focus the research with more clarity.

  • Introduction: The objective is not clear and it does not refer to Ecuador.

Aspects and references have been included that help to specify this research in relation to Ecuador.

  • Line 97: “In this case, in search of better alternatives, the AHP method developed by Thomas Saaty [18].” This sentence seems to be incomplete or their meaning is not really clear.

This defect has been corrected.

  • It would be also desirable to delve into the theoretical aspects. Section 2 (Theoretical framework) does not address the subject in depth.

In the revised version of the manuscript, the theoretical contents of the research have been deepened.

  • Line 110 to 115 explain the methodology used to study the state of the art. Two comments: a) According to the results, this process was not exhaustive enough, as it is said previously; b) It is not very usual to explain the methodology of consulting the database of bibliographic references used to study the theoretical framework, since it is common knowledge to all scientific work.

These issues are corrected and clarified in the revised version as will prove

  • Line 116 to 118: Why it is analyzed only the literature of the last five years? In spite of the fact that it is important to include current references, to encompass the state of the art, time framework should not be delimited. Especially, since it is later said that this time framework is exceeded on some occasions, without the reason being sufficiently justified.

These issues are corrected and clarified in the revised version as will prove

  • Although the paper cites 86 references, wich shows a vast knowledge of the topic, there are too many references to web pages and some references are a little general and not so important to your research. It could also be deepened in other relevant English-speaking references that are missing, according to what it was said before.

Dear reviewer, thank you very much for your comments, no doubt they improve the manuscript, its rigor and depth. Then I will reply one by one to your comments.
In turn, as you can see, other relevant references have been included for the investigation.

Kind Regards and happy new year

Prof. Dr. Ramón Rueda López

Round 2

Reviewer 1 Report

The paper is improved. I do not have further queries.